# A network modelling approach to assess non-pharmaceutical disease controls in a worker population: An application to SARS-CoV-2

**Edward M. Hill**[1,2]⊙*, **Benjamin D. Atkins**[1,2]⊙, **Matt J. Keeling**[1,2], **Louise Dyson**[1,2], **Michael J. Tildesley**[1,2]

**1** The Zeeman Institute for Systems Biology & Infectious Disease Epidemiology Research, School of Life Sciences and Mathematics Institute, University of Warwick, Coventry, United Kingdom, **2** Joint UNIversities Pandemic and Epidemiological Research, https://maths.org/juniper/

⊙ These authors contributed equally to this work.
* Edward.Hill@warwick.ac.uk

**Data Availability Statement:** The Social Contact Survey data are available from https://wrap.warwick.ac.uk/54273/. All other data utilised in this study are publicly available, with relevant

## Abstract

As part of a concerted pandemic response to protect public health, businesses can enact non-pharmaceutical controls to minimise exposure to pathogens in workplaces and premises open to the public. Amendments to working practices can lead to the amount, duration and/or proximity of interactions being changed, ultimately altering the dynamics of disease spread. These modifications could be specific to the type of business being operated. We use a data-driven approach to parameterise an individual-based network model for transmission of SARS-CoV-2 amongst the working population, stratified into work sectors. The network is comprised of layered contacts to consider the risk of spread in multiple encounter settings (workplaces, households, social and other). We analyse several interventions targeted towards working practices: mandating a fraction of the population to work from home; using temporally asynchronous work patterns; and introducing measures to create 'COVID-secure' workplaces. We also assess the general role of adherence to (or effectiveness of) isolation and test and trace measures and demonstrate the impact of all these interventions across a variety of relevant metrics. The progress of the epidemic can be significantly hindered by instructing a significant proportion of the workforce to work from home. Furthermore, if required to be present at the workplace, asynchronous work patterns can help to reduce infections when compared with scenarios where all workers work on the same days, particularly for longer working weeks. When assessing COVID-secure workplace measures, we found that smaller work teams and a greater reduction in transmission risk reduced the probability of large, prolonged outbreaks. Finally, following isolation guidance and engaging with contact tracing without other measures is an effective tool to curb transmission, but is highly sensitive to adherence levels. In the absence of sufficient adherence to non-pharmaceutical interventions, our results indicate a high likelihood of SARS-CoV-2 spreading widely throughout a worker population. Given the heterogeneity of demographic attributes across worker roles, in addition to the individual nature of controls such as contact tracing, we demonstrate the utility of a network model approach

references and data repositories stated within the main manuscript and Supporting information. The code repository for the study is available at: https://github.com/EdMHill/covid19_worker_network_model.

**Funding:** EMH, BDA, MJK, LD and MJT were supported by the Medical Research Council through the COVID-19 Rapid Response Rolling Call [grant number MR/V009761/1]; MJK, LD and MJT were supported by the Engineering and Physical Sciences Research Council through the MathSys CDT [grant number EP/S022244/1]; MJK, LD and MJT were supported by UKRI through the JUNIPER modelling consortium [grant number MR/V038613/1]. The funders had no role in study design, data collection and analysis, decision to publish, or preparation of the manuscript.

**Competing interests:** The authors have declared that no competing interests exist.

to investigate workplace-targeted intervention strategies and the role of test, trace and isolation in tackling disease spread.

## Author summary

As part of a collective effort to protect public health by disrupting viral transmission of SARS-CoV-2, businesses have implemented measures to minimise exposure to coronavirus in workplaces and premises open to the public. Adjustments in working practices can result in changes to patterns of interaction, altering the dynamics of viral spread. To assess the impact of workplace targeted non-pharmaceutical disease controls against epidemic spread of SARS-CoV-2 amongst a population of workers, we present a network-based model with layered contacts capturing multiple encounter settings (workplaces, households, social and other). Informed by UK data, the model accounts for work sector, workplace size and the division of time between work and home. We study three workplace focused interventions: (i) a specified fraction of each work sector working from home; (ii) temporally asynchronous work patterns; (iii) introduction of COVID-secure workplaces. We also examine the role of adherence to isolation and test and trace measures. Following isolation guidance and engaging with contact tracing alone is an effective tool to curb transmission, but is highly sensitive to adherence levels. Given the heterogeneity of demographic attributes across worker roles, we demonstrate the utility of a network model approach to investigate workplace-targeted control measures against infectious disease spread.

## Introduction

Globally, many countries have employed social distancing measures and non-pharmaceutical interventions (NPIs) to curb the spread of SARS-CoV-2 [1]. For many individuals, infection develops into COVID-19 disease, with symptoms including fever, shortness of breath, and altered sense of taste and smell, potentially escalating to a more severe state which may include pneumonia, sepsis, and kidney failure [2]. In the United Kingdom (UK), the enaction of lockdown on 23rd March 2020 saw the closure of workplaces, pubs, restaurants and the restriction of a range of leisure activities. As the number of daily confirmed cases went into decline during April, May and into June [3], measures to ease lockdown restrictions began; some non-essential businesses were permitted to re-open and small groups of individuals from different households were allowed to meet up outdoors, whilst maintaining social distancing.

By the end of September 2020, exponential growth had returned in almost all regions of the UK [4, 5] and stricter controls were subsequently reintroduced to curtail growth. Whilst lockdown has been a strategy used around the world to reduce the public health impacts of COVID-19, it is important to recognise that such strategies are very disruptive to multiple elements of society [6, 7], especially given that restrictions are largely unpredictable to local populations and businesses.

As part of a collective effort to protect public health by disrupting viral transmission, businesses also need to act appropriately by taking all reasonable measures to minimise exposure to coronavirus in workplaces and premises open to the public. In the UK, each of the four nations (England, Wales, Scotland, Northern Ireland) has published guidance to help employers, employees and the self-employed to work safely [8–11]. Adjustments in working practices

can result in changes to the amount, duration, and/or proximity of interactions, thereby altering the dynamics of viral spread. These modifications could be variable depending upon the type of business being operated and may include limiting the number of workers attending a workplace on any given day, as well as introducing measures to make a workplace COVID-secure, such as compulsory mask wearing and the use of screens. For this paper, we are interested in how interventions targeting workplace practices may affect infectious disease control efforts, whilst accounting for the variation in employee demographics across working sectors.

Modelling has been contributing to the COVID-19 pandemic response, with analyses having been carried out pertaining to transmission of SARS-CoV-2 within specific parts of society, including health care workers [12], care homes [13], university students [14–16], and school pupils and staff [17–19]. As in these studies, we view the contact structure for the adult workforce as being comprised of several distinct layers. Knowledge of the contact structure allows models to compute the epidemic dynamics at the population scale from the individual-level behaviour of infections [20]. More generally, such models of infectious disease transmission are a tool that can be used to assess the impact of options seeking to control a disease outbreak.

In this study, we outline an individual-based network model for transmission of SARS-CoV-2 amongst the working population. Informed by UK data, the model takes into account work sector, workplace size and the division of time between work and home. In addition to workplace interactions, contacts also occur in household and social settings. Given the heterogeneity of demographic attributes across worker roles, as well as the use of individual-based NPIs such as contact tracing, we demonstrate the utility of a network model approach in investigating workplace-targeted control measures against infectious disease spread.

## Methods

We simulated an epidemic process over a network of workers and assessed the impact of workplace-targeted non-pharmaceutical interventions. In this section we detail: (i) the structure of the network model, (ii) the data sources used to parameterise the network contact structure, (iii) the model for SARS-CoV-2 transmission and COVID-19 disease progression, and (iv) the simulation protocol used to assess the scenarios of interest.

### Network model description

Each node in the network represented a worker. We did not include in the network children, the elderly, or working-age individuals not in employment (this is an acknowledged limitation of the system). The entire network was assumed to be contained within the same geographical area, such that any node could possibly be linked with any other node. We used a multi-layered network model to capture common contact settings. Our model was comprised of four layers: (i) households, (ii) workplaces, (iii) social contacts, and (iv) other contacts.

**Household contact layer.** To allocate workers to households, we sampled from an empirical distribution based on data from the 2011 census in England [21]. To obtain this distribution, we calculated the proportion of households containing 1 to 6+ people between the ages of 20–70 (Fig H in the S1 Text). Thus, as previously highlighted, we omit children and the elderly from our analysis. When sampling from this distribution, we restricted the maximum household size to six people. Since we assumed that everyone in a household is an active worker, this size restriction helped to reduce the overestimation of the number of active workers mixing within households. Within each household, members formed fully connected networks.

**Workplace contact layer.** To disaggregate working sectors, we used data from the 2020 edition of the ONS 'UK business: activity, size and location database' [22]. Specifically, we took counts (for the UK) of the number of workplaces, stratified into 88 industry divisions/615

**Table 1. Within the network model, workplaces were grouped into the following 41 industrial sectors.**

| | | | |
|---|---|---|---|
| 1. Agriculture | 12. Postal | 23. Employment and HR | 34. Betting |
| 2. Mining | 13. Accommodation | 24. Travel Agency | 35. Sport |
| 3. Manufacturing (food & beverages) | 14. Restaurant and Bar | 25. Security | 36. Theme Parks |
| 4. Manufacturing (other) | 15. Broadcasting and Communications | 26. Cleaning | 37. Religious and Political Organisations |
| 5. Utilities and Waste | 16. Information Technology | 27. Office (other) | 38. Repair |
| 6. Construction | 17. News | 28. Public Administration and Defence | 39. Hairdressers |
| 7. Motor Trade | 18. Banking/Accounting | 29. Education | 40. Funeral |
| 8. Wholesale | 19. Real Estate | 30. Hospital/Doctor/Dental | 41. Personal Services |
| 9. Retail | 20. Professional/Science/Tech | 31. Care Homes | |
| 10. Transport | 21. Veterinary | 32. Social Work | |
| 11. Transport Support | 22. Rental Companies | 33. Arts | |

industry classes (Standard Industrial Classifications (UK SIC2007)) and workforce sizes (0–4, 5–9, 10–19, 20–49, 50–99, 100–249, 250+).

We reassigned the industry types to one of 41 sectors (see Table 1 for a listing of the work sectors). We generated a set of workplaces and workplace sizes for each of the 41 sectors in a two-step process: first, we sampled from the relevant empirical cumulative distribution function of the binned workplace size data to obtain the relevant range. For a bounded range (all but the largest bin), we then sampled an integer value according to a uniform distribution that spanned the selected range. Since the largest data bin (250+ employees) is unbounded, in this instance we instead sampled from a shifted Gamma(1,100) distribution (shape and scale parameterisation, shifted to 250). When sampling the number of workplaces and individual workplace sizes in each simulation run, there was variation in these distributions, though qualitative features were retained in individual realisations (Fig I in the S1 Text).

We separated workplace contacts into static contacts and dynamic contacts. For static contacts, we constructed the network to allow for contacts both within a worker's workplace (most common) and to other workplaces in the same industrial sector (less common). These contacts occurred every workday, unless either person was working from home, and remained unchanged throughout the simulation. We generated static contacts using a 'configuration model' style algorithm, allowing the specification of a desired degree distribution for each sector. We adapted the standard configuration model to allow a variable amount of clustering, where a higher value of clustering led to more contacts being made within a workplace compared to between different workplaces. We subjectively assumed throughout that the probability of making contact with an individual in another workplace, compared to an individual within the same workplace, was 0.05. We applied this consistently across sectors as a simplifying assumption, however, were relevant fine-scale data available then sector-specific parameterisations may be more appropriate. Unlike the standard configuration model, we did not allow edges to be made with oneself or repeated edges. As such, the resulting degree distribution was an approximation of the distribution used as an input. For the steps defining the algorithm, see Section A.1 in the S1 Text.

Dynamic contacts represented those that may occur between workers and non-workers, though still in the workplace, for example contacts between retail workers and shoppers. These were regenerated every day: for each worker (not working from home), we generated a number of dynamic contacts from a sector-specific degree distribution and assigned the recipients at random. These were not clustered in any way; that is, every person in the population had an equal probability of being the recipient, though we did not allow repeated edges or edges with

oneself. Given that the number of dynamic contacts per person is small compared to the size of the population, the desired degree distribution was approximately preserved.

**Social contacts.**   Social contacts were generated in two stages. First, we generated a 'social group' for each person. We used a similar configuration model style algorithm as for the generation of static workplace contacts, allowing the specification of a desired degree distribution. We adapted the standard configuration model to allow for greater clustering (which in this context relates to the probability that each contact is made with a friend-of-a-friend, opposed to someone at random, set at 0.5) and did not allow edges with oneself or repeated edges. This resulted in an acceptable approximation of the desired degree distribution. The second step specified who a person socialised with each day: for each individual on each day, we sampled a subset of their social group to construct the social contacts made on that day. The number of social contacts made per day was specified by a degree distribution (but restricted by the size of their social group), which we allowed to differ between workdays and non-workdays. We provide a description of the steps for constructing the social contact layer of the network in Section A.2 of the S1 Text.

**Other contacts.**   The final contact layer captured random, dynamic, contacts made each day with any other individuals in the population (for example on public transport). We used a fixed daily probability of each individual interacting with any other individual in the network, without clustering or preference. We justify this with the assumption that the entire network is contained within the same geographical area.

## Contact parameterisation

We parameterised the number of contacts that occurred within each layer of the network using data from the University of Warwick Social Contact Survey [23–25]. The Social Contact Survey was a paper-based and online survey of 5,388 participants in the United Kingdom conducted in 2010. We extracted records provided by 1,860 participants, with a total of 34,004 contacts (eligibility criteria are outlined in Section B of the S1 Text). This data informed the network construction parameters for the workplace and social layers, with stratification according to sector. We fit parameters for these contact distributions using maximum likelihood estimation via the FITDISTRPLUS package in R. We present a summary of network parameters in Table 2.

**Workplace contacts.**   We used the Warwick Social Contact Survey to parameterise the degree distributions for both static and dynamic contacts occurring in workplaces. For a full description of the workplace contact layer parameterisation, including the mapping between the ONS sectors and occupations listed in the Contact Survey, see Section B.1 in the S1 Text.

**Table 2. Description of network contact parameters.** Lognormal distributions are described using a mean and standard deviation parameterisation. All values are given to 2 decimal places.

| Description | Degree distribution | Source |
|---|---|---|
| $N$, network size | 10,000 | Assumption |
| Household (static) | Fully connected | Assumption |
| Work setting | See Section B.1 in the S1 Text | Fitted from Social Contact Survey [23–25] |
| Friendship group size | Lognormal(3.14,1.141) | Fitted from Social Contact Survey [23–25] |
| Social contacts (workday) | Lognormal(1.40,1.27) | Fitted from Social Contact Survey [23–25] |
| Social contacts (non-workday) | Lognormal(1.54,1.15) | Fitted from Social Contact Survey [23–25] |
| Other contact probability | $\frac{1}{N}$ | Assumption |
| Between workplace contact probability | 0.05 | Assumption |
| Friend-of-friend probability | 0.5 | Assumption |

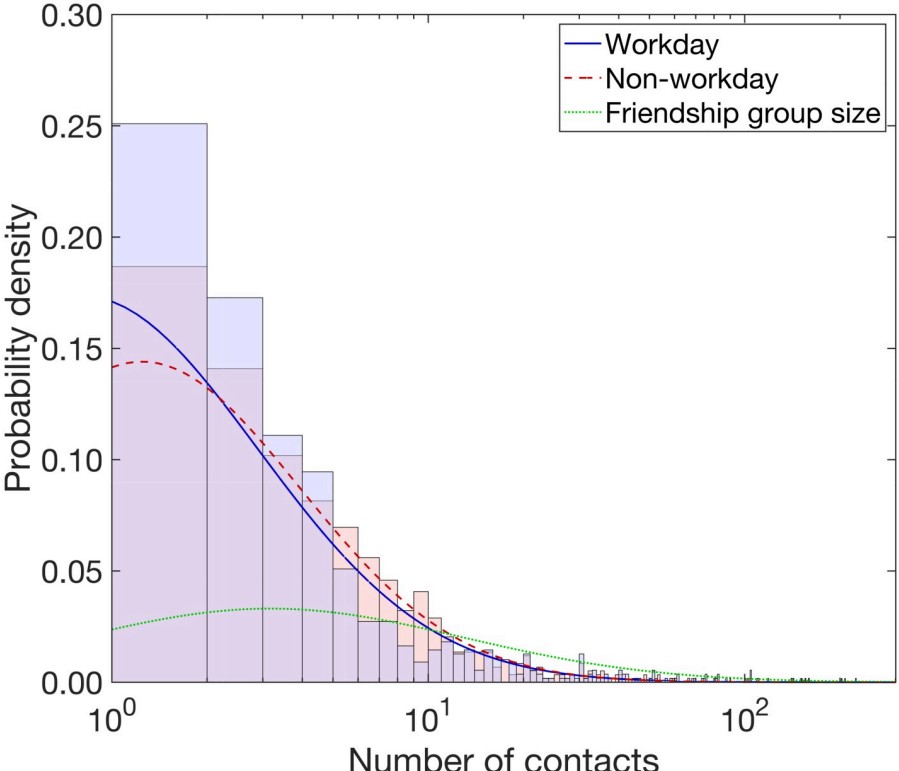

**Fig 1. Density functions of the best fit lognormal distributions with respect to number of social contacts each day and friendship group size.** All traces are plotted against the number of contacts, presented on a log scale. The stated lognormal (LN) distributions that follow are given using a mean and standard deviation parameterisation. We obtained a heavier tailed distribution for number of social contacts per day on non-workdays (red dashed line, LN (1.54, 1.15)) versus workdays (blue solid line, LN(1.40, 1.27)). The histograms show the associated empirical probability densities in a matching colour scheme. Friendship group sizes were sampled from a LN(3.14, 1.41) distribution, depicted by the green dotted line.

We found that, across all work sectors, the daily number of workplace contacts displayed a heavy tail. Thus, we chose to fit lognormal (LN) distributions to the data, which consistently provided stronger correspondence to the data, across different occupations, than alternative choices of distribution.

**Social contacts.** We used data from the Warwick Social Contact Survey to acquire a distribution of social group sizes and estimate the daily number of social contacts on both work and non-work days. To acquire a distribution of social group sizes, we scaled up the contacts recorded in the Warwick Social Contact Survey, resulting in a LN(3.14, 1.41) distribution with a mean and standard deviation parameterisation (Fig 1 and Table 2, full methodological details in Section B.2 of the S1 Text).

Through fitting distributions to the workday and non-workday data independently, we obtained LN(1.40, 1.27) and LN(1.54, 1.15) distributions for workday and non-workday social contacts respectively (Fig 1 and Table 2, for additional information see Section B.3 in the S1 Text).

**Other contacts.** To capture other miscellaneous, randomly-occurring contacts, for each individual on each day, we generated random contacts according to a fixed probability. We set this probability so that each individual had, on average, one additional contact per day (Table 2), resulting in a Poisson(1) distribution across the entire population.

## Epidemiological model

**Disease states.**   We ran a disease process on the network structure, with each individual being in either a susceptible, latent (infected but not infectious), infectious or recovered state.

Once infected, we assumed infectiousness could start from the following day. We assumed an Erlang-distributed incubation period, with shape parameter 6 and scale parameter 0.88 [26].

The distribution of infectiousness had a four day pre-symptomatic phase, followed by a ten day symptomatic phase. This gave a total of 14 days of infectivity and a minimum 15 day infection duration. The infectiousness temporal profile weighted the contact setting transmission risk (see the subsequent subsection on *Setting transmission risk*) across the duration of the infectious period (for the full temporal profile, see Table 3). It was based on a Gamma(97.2, 0.2689) distribution, with shape and scale parameterisation, shifted by 25.6 days [27, 28]. Following completion of the infectious period, the individual entered the recovered state.

**Asymptomatic transmission.**   Infected individuals could be either asymptomatic or symptomatic, according to a specified asymptomatic probability. There remains significant uncertainty as to what this probability should be, however community surveillance studies informed this parameter. The REal-time Assessment of Community Transmission-1 (REACT-1) study found approximately 70% of swab-positive adults and 80% of swab-positive children were asymptomatic at the time of swab and in the week prior [3]. Note that this includes pre-symptomatic infected individuals who would later go on to display symptoms. This fell to 50% at later stages of the study [5]. To reflect this uncertainty, for each simulation we sampled the asymptomatic probability from a Uniform(0.5, 0.8) distribution.

There remains limited data available to provide a robust quantitative estimate of the relative infectiousness of asymptomatic and symptomatic individuals. However, there are indications that asymptomatic individuals could be less infectious than symptomatic individuals [29, 30]. Therefore, we assumed that asymptomatic individuals had a lower risk of transmitting infection compared to symptomatic individuals. To reflect the uncertainty in this area, for each simulation we sampled the relative infectiousness of asymptomatics compared to symptomatics from a Uniform(0.3, 0.7) distribution. This was sampled independently to the asymptomatic probability. The sampled value was applied as a scaling on transmission risk, applied evenly throughout the duration of infectiousness (*i.e.* with no time dependence, see the subsection *Probability of transmission per contact*).

**Setting transmission risk.**   Attributing risk of transmission to any particular contact in a particular setting is complex, due to the huge heterogeneity in contact types. We used a data-driven approach to obtain the relative risk of transmission within each network layer. We then scaled these risks equally in order to obtain an appropriate growth rate of the disease.

For each contact setting (network layer), the transmission risk corresponded to the probability of a susceptible individual being infected over the course of the entire infectious period

**Table 3. Description of epidemiological parameters.**

| Description | Distribution | Source |
|---|---|---|
| Incubation period | Erlang(6, 0.88) | [26] |
| Infectiousness profile | Infectivity profile over 14 days: [0.0369, 0.0491, 0.0835, 0.1190, 0.1439, 0.1497, 0.1354, 0.1076, 0.0757, 0.0476, 0.0269, 0.0138, 0.0064, 0.0044] | [27, 28] |
| Proportion of cases asymptomatic | Uniform(0.5, 0.8) | REACT-1 study [3, 5] |
| Relative infectiousness of an asymptomatic | Uniform(0.3, 0.7) | [29, 30] |

for an infected with a relative infectiousness of 1, assuming the susceptible and infectious individual were in contact in the specified setting every day. The transmission risk was then scaled to obtain the probability of transmission occurring across a susceptible-infectious contact pair on a given day (see the subsection *Probability of transmission per contact*).

For household transmission, we used estimates of adjusted household secondary attack rates from a UK based surveillance study [31]. We attributed a household secondary attack rate to each student based on the size of their household. We sampled the attack rates from a normal distribution with mean dependent on the household size: 0.48 for a household size of two, 0.40 for three, 0.33 for four, and 0.22 for five or more. The standard deviation of the normal distribution for households of size two or three was 0.06, and for households of four or more was 0.05. We highlight that these estimates were made using a sample of 379 confirmed COVID-19 cases, meaning the robustness of the central estimates could be low. As such, we sampled from a distribution to ensure this uncertainty was captured.

For transmission risk in other settings, we performed a mapping from the Social Contact Survey [23–25] to obtain a relative transmission risk compared to household transmission. To obtain the means, we used the central estimate of adjusted household secondary attack rate for those aged 18–34 of 0.34 [31] and scaled this based on the characteristics of contacts in different locations, obtained from the contact survey (further details in Section C of the S1 Text). Standard deviations were set to have a constant size relative to the mean. Transmission risks were consistent across all non-household settings, except within organised societies where we assigned a lower transmission risk to reflect the implementation of COVID-secure measures that would be required to permit these meetings to take place.

We calibrated the relative transmission risks to achieve an uncontrolled reproductive number, $R_t$, that was, on average, in the range 2–4 for the initial phase of the outbreak. To obtain these early phase transmission dynamics, we applied an equal scaling of 0.8 to all of the transmission risks calculated above (see Section D in the S1 Text).

**Probability of transmission per contact.**   We outline here how the setting transmission risk, infectiousness temporal profile and relative infectiousness of an individual were used to compute the probability of transmission across an infectious-susceptible contact pair.

For an infectious individual $j$ on day $t$ of their infectious state, the probability of transmission per susceptible contact in contact setting $s$, denoted $p_{j,s}(t)$, was given by the product of four components:

$$p_{j,s}(t) = r_s \times i(t) \times a_j \times 0.8,$$

with $r_s$ the transmission risk in setting $s$, $i(t)$ the value of the infectiousness temporal profile on day $t$ (Table 3), $a_j$ the relative infectiousness of individual $j$ (taking either value 1 for cases that were symptomatic during the infection episode, or the sampled value for relative infectiousness of asymptomatics compared to symptomatics otherwise), and 0.8 the scaling applied to calibrate the system to achieve (in the majority of simulations) an $R_t$ in the range of 2–4 for the initial phase of the outbreak.

## Isolation, test and trace

**Testing and isolation measures.**   Upon symptom onset, an adhering individual would immediately take a test and enter isolation for 10 days. At that time, their household would also enter self-isolation for 14 days (matching the UK government guidance prior to 14th December 2020, when self-isolation for contacts of people with confirmed coronavirus was shortened from 14 days to 10 days across the UK) [32]. Isolation was assumed to remove all non-household contacts for the period of isolation.

We assumed that an isolating individual would remain in isolation for the required amount of time, or until a negative test result was returned. We included a two day delay between taking the test and receiving the result. We assumed the test had 100% specificity and its sensitivity was dependent upon time since infection (we used the posterior median profile of the probability of detecting infection reported by Hellewell *et al.* [33]).

In the event that the test result from the index case was negative, household members would be released from isolation, as long as no other members had become symptomatic during that time. The index case remained in self-isolation if they had independently been identified via contact tracing as a contact of a known infected; otherwise, that individual also left self-isolation.

**Forward contact tracing.** The modelled tracing scheme looked up contacts for an index case up to five days in the past. It was assumed that tracing took place on the third day after symptom onset, following testing and a two day delay to return a positive result. Thus contacts may be recalled up to two days prior to the onset of symptoms. We assumed that an individual would be able to recall all of their regular contacts for that time. However, we assumed that the probability of being able to recall their 'dynamic' contacts diminished with time, from 0.5 one day previously, reducing in increments of 0.1, such that the probability of successfully tracing a contact five days in the past is 0.1. Other assumptions could be explored and a wider range of assumptions, collectively, would generate more variation in the results.

Contacts of a confirmed case were required to spend up to 14 days in self-isolation [34] (matching the UK government guidance prior to 14th December 2020, when self-isolation for contacts of people with confirmed coronavirus was shortened from 14 days to 10 days across the UK). We set the isolation period to elapse 14 days after the index case became symptomatic.

**Adherence.** We used an adherence parameter to capture the proportion of individuals that follow the recommended guidance. This was applied to the isolation and test-and-trace measures, representing the probability that an individual would both adhere to isolation guidance and engage with test and trace. We did not allow partial adherence to one measure and not the other. Adherence to isolation encompassed isolation for any reason: presenting with symptoms themselves, being in the same household as someone displaying symptoms, or being identified as a close contact of an infected individual via contact tracing. Unless otherwise stated, we assumed a constant 70% adherence to isolation and test-and-trace measures.

We give an overview of isolation, test and trace related parameters in Table 4.

**Table 4. Description of isolation, test and trace related parameters.**

| Description | Value | Source |
|---|---|---|
| Duration of self-isolation if symptomatic | 10 days | UK government guidance [32] |
| Household isolation period | 14 days | UK government guidance (prior to 14th December 2020) [32] |
| Duration of isolation if contact traced | 14 days (beginning from the day the index case first displays symptoms) | UK government guidance (prior to 14th December 2020) [34] |
| Delay in receiving test result | 2 days | Assumption |
| Dynamic contact recall | For five previous days, [0.5, 0.4, 0.3, 0.2, 0.1]. Zero probability beyond five days. | Assumption |
| Adherence | 0.7 | Assumption |

## Simulation outline

We used this model framework to evaluate the transmission dynamics of SARS-CoV-2 amongst the workforce under different workplace-targeted NPIs. We also assessed the role of adherence to the underlying social distancing guidance and engagement with test-and-trace.

We ran all simulations with a population of 10,000 workers and a simulation time corresponding to 365 days. The size of the network was kept small due to the assumption that any node could contact any other, thus must be in the same geographical area. For the default working pattern, we applied a simplifying assumption that all workers had the same working pattern of five days at the workplace (that can be considered to be Monday to Friday) and two days off (Saturday and Sunday). This applied unless otherwise stated. Ten individuals began the simulations in an infectious state, of whom between 5–8 were asymptomatic (randomly sampled) and the remaining symptomatic (between 2–5 individuals). All other individuals began the simulations in a susceptible state.

Unless stated otherwise, we assumed that all NPIs, including isolation and test-and-trace, were implemented from day 15. For the two weeks prior to this, the virus was assumed to spread unhindered. Once isolation guidance began, any pre-existing symptomatic, adherent individuals would follow the new guidance, entering isolation themselves until 10 days after symptom onset. Adhering household members would also enter isolation until 14 days after symptom onset for the index case. However, only those that developed symptoms after the implementation of test-and-trace would be tested and contact traced.

For each parameter configuration, we ran 1,000 simulations, amalgamating 50 batches of 20 replicates. Each batch of 20 replicates was obtained using a distinct network realisation. We performed the model simulations in Julia v1.5. The code repository for the study is available at: https://github.com/EdMHill/covid19_worker_network_model.

Our assessment comprised of four strands, assessing the impact of: i) a proportion of workers working from home; ii) different working patterns; iii) the introduction of COVID-secure workplace measures; and iv) the level of adherence to isolation and test-and-trace interventions. We detail these below. Across all sections of analysis, we primarily focused on measures associated with outbreak severity (size and peak in infectious case prevalence), outbreak duration, and extent of isolation (cumulative isolation time). Further measures are provided in the S1 Text.

**Proportion of the workforce working from home.**   We investigated the impact of different proportions of the workforce working from home full time. We varied this proportion (consistent across all sectors) from 0 to 1 in increments of 0.1. We also included a scenario in which the proportion of workers working from home was not consistent across work sectors (summarised in Table 5). For this, we subjectively set the proportion of workers working from home to be highest in office based roles (70% working from home), at a moderate level in primary and manufacturing trade occupations (for example, repair with 50% working from home and construction with 30% working from home), lower in sales and customer service roles such as retail (20% working from home) and zero for those in the education, health, care home and social work sectors (0% working from home). Overall for this scenario, approximately 35% of the workforce was working from home.

**Worker patterns.**   We explored two alternative choices related to the scheduling of workers being present at their usual workplace: (i) synchronous work pattern—workers returned to work for a given number of days per week (between Monday to Friday inclusive), with all workers scheduled to work on the same days; or (ii) asynchronous working pattern—workers returned to work for a given number of days per week, with the days of return randomly assigned to each worker (between Monday to Sunday inclusive).

**Table 5. Sector-specific working from home proportions.**

| Sector | Prop. | Sector | Prop. | Sector | Prop. |
|---|---|---|---|---|---|
| Agriculture | 0.3 | Broadcasting/Comm. | 0.7 | Education | 0 |
| Mining | 0.3 | IT | 0.7 | Hospital/Doctor/Dental | 0 |
| Manufact. (food) | 0.3 | News | 0.7 | Care Homes | 0 |
| Manufact. (other) | 0.3 | Banking/Accounting | 0.7 | Social Work | 0 |
| Util. and Waste | 0.3 | Real Estate | 0.7 | Arts | 0.2 |
| Construction | 0.3 | Professional/Sci/Tech | 0.7 | Betting | 0.2 |
| Motor Trade | 0.5 | Vet | 0.2 | Sports | 0.2 |
| Wholesale | 0.2 | Rental Company | 0.7 | Theme Parks | 0.3 |
| Retail | 0.2 | Employment/ HR | 0.7 | Religious Org. | 0.7 |
| Transport | 0.5 | Travel Agency | 0.7 | Repair | 0.5 |
| Transport Support | 0.5 | Security | 0.3 | Hairdressers | 0.2 |
| Postal | 0.2 | Cleaning | 0.5 | Funerals | 0.2 |
| Accomm. | 0.2 | Office | 0.7 | Personal Services | 0.7 |
| Restaurant/Bar | 0.2 | Public/Admin/Defence | 0.7 | | |

**COVID-secure workplaces.**   We defined a workplace to be 'COVID-secure' if measures had been taken to reduce the number of contacts workers had and decrease the risk of transmission for those contacts that remained. We assessed the impact of all workplaces undergoing changes to their contact structures, combined with a possible reduction in transmission risk across work based contacts. We simulated all combinations of work team sizes of 2, 5 or 10, in conjunction with the scaling of the baseline work sector transmission risks (for both static and dynamic work contacts) by a factor of either 0.25, 0.5, 0.75 or 1. We assumed that everyone within a team was connected with each other, but with no one else at the workplace. No regular work contacts were made outside a worker's workplace. We did not amend the distributions of dynamic contacts occurring at the workplace.

As well as the baseline assumption of 70% adherence to isolation and test-and-trace measures, in order to highlight the effects brought about solely by COVID-secure measures (in the absence of other NPIs), we also ran simulations with 0% adherence (*i.e.* in the absence of) and 100% adherence to isolation and test-and-trace measures.

**Adherence to isolation, test and trace.**   Finally, we analysed the sensitivity of our model to the underlying adherence parameter, which defines whether or not an individual will both adhere to isolation guidelines and engage with test-and-trace. We sampled adherence between 0 and 1 in increments of 0.1. To conclude, we explored the sensitivity to the adherence parameter of the impact of workplace interventions on outbreak severity, outbreak duration, and extent of isolation.

## Results

### Working from home

We found that a greater proportion of the workforce working from home was effective in reducing the final size of the outbreak, the peak in infectious cases, and total-isolation-days (Fig 2(a)–2(c)). An increase from no one working from home (corresponding to there being no changes in the work pattern policy) to everyone working from home resulted in a 60–70% decrease in the medians of each of these metrics. However, working from home was relatively ineffective in reducing outbreak duration and displayed a non-monotonic relationship (Fig 2(d)). For increases in the proportion working from home between 0–70%, we observed an

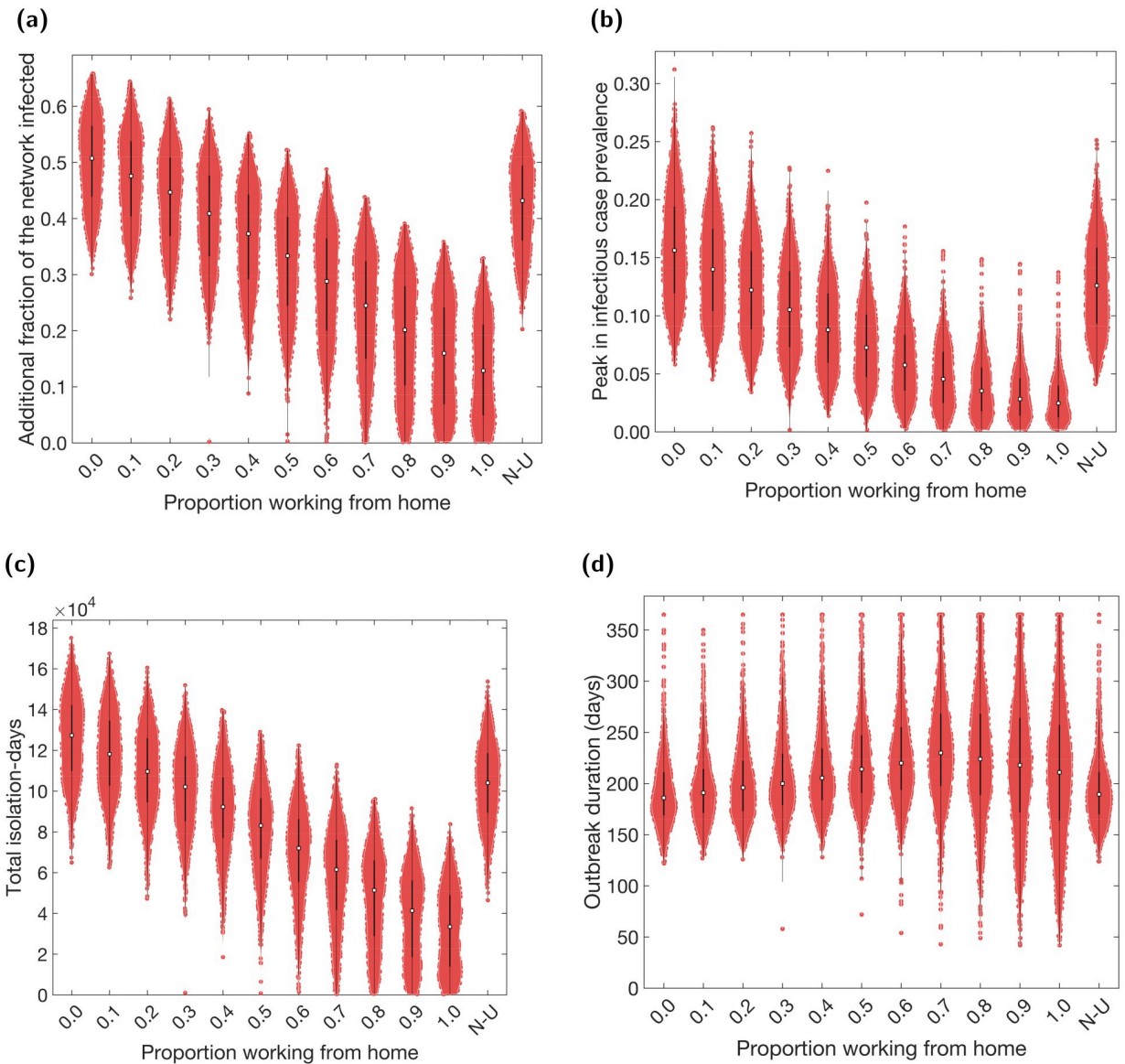

**Fig 2. Case and isolation summary statistics under differing fractions of workers working from home.** We introduced NPIs from day 15 onwards, with varying proportions of the workforce working from home. N-U corresponds to non-uniform proportions working from home across the work sectors (see Table 5). Outputs are summarised from 1,000 simulations (20 runs per network for 50 separate network realisations). We assumed an adherence of 70% in all runs. The white markers denote medians and solid black lines span the 25th to 75th percentiles. We give central and 95% prediction intervals in Table D in the S1 Text. **(a)** Additional proportion of the population that were infectious post introduction of NPIs (day 15 onwards). **(b)** Peak in infectious case prevalence. **(c)** Total isolation-days. **(d)** Outbreak duration (days).

increase in the median outbreak duration from 186 days (95% prediction interval (PI): 140–272 days) to 230 days (95% PI: 128–365 days). Further increases in the proportion working from home between 70–100% resulted in a decrease in the median duration, reaching 211 days (95% PI: 81–365 days). Finally, we observed a consistent increase in variability in duration across simulations as more people worked from home.

The relationships observed in Fig 2 can also be seen in the temporal profiles of the proportion of the population infectious, the proportion isolating, and $R_t$ (left column of Fig J in the S1 Text). A greater proportion of the workforce working from home resulted in a faster decrease

in $R_t$ towards 1 during the early stages of the outbreak. This resulted in a flattened epidemic curve, observed in both infection and isolation levels. However, in the long run, $R_t$ remained marginally higher (though below 1), due to a larger susceptible population. This allowed the outbreaks to last longer.

Thus far, we have assumed that the proportion working from home applies equally across all industry sectors. However, in reality, such an approach may not be implementable, due to the differing nature of such sectors. We demonstrated the flexibility of the model construction by also simulating one example of a scenario in which the proportion working from home was sector-dependent (labelled N-U in Fig 2). The sector-dependent proportions used resulted in approximately 35% of the total population of workers working from home. However, when compared to the results using an equal proportion across all sectors, the non-uniform simulations appeared closer to the results obtained from a proportion working from home lower than 35% (10–25% depending on the metric used). Thus, correlation between the amount of contacts workers within a sector have with the general public and the ability of those workers to work from home may reduce the effectiveness of a work from home policy.

We note that there was significant variation in infection and isolation outcomes for each working from home intervention scenario. This was primarily due to the variability in epidemiological factors between simulation runs, such as the distribution of initial infections, the asymptomatic probability, and the relative infectiousness of an asymptomatic case, all of which were randomly generated at the start of each simulation. The different network structures contributed relatively less variation. Results from a collection of simulations performed on a single network realisation also display a large amount of variation (Fig K in the S1 Text), a characteristic observed for each form of intervention studied in subsequent result subsections (not shown).

## Worker patterns

Rather than stipulating a proportion of the population to work from home full-time (five days a week), we can instead consider the case where workers only work from home on specified days and are physically present at their workplace otherwise. We varied the number of days spent at the workplace from 0 to 5 and considered working schedules that were: i) synchronous: workers returned to work for the given number of days per week, with all workers scheduled to work on the same days (between Monday and Friday inclusive); or ii) asynchronous: workers returned to work for the given number of days per week, with the days of return randomly assigned to each worker (between Monday and Sunday inclusive).

The number of days workers spend at the workplace had a similar effect on the reported metrics as the proportion working from home. Fewer days at the workplace resulted in fewer infections overall (Fig 3(a)), a lower peak in infectious case prevalence (Fig 3(b)) and fewer isolating individuals (Fig 3(c)). However, there was relatively less effect on the outbreak duration and the relationship is non-monotonic (Fig 3(d)). These relationships were again displayed in the temporal profiles of the proportion of people infectious, isolating, and $R_t$ (Fig J in S1 Text, centre and right columns).

Compared to a work from home policy, in which a proportion of workers work from home all the time, allowing all workers to work from home some of the time was less effective if work patterns were synchronous, but more effective if they were asynchronous. For example, comparing a policy of 40% of workers working from home all the time (Fig 2) to a policy of all workers spending 40% of their time working from home (two days per week working from home, three days per week at the workplace; Fig 3), we found the former resulted in a median of 37% (95% PI: 19%–51%) of the population infected (post-introduction of interventions on

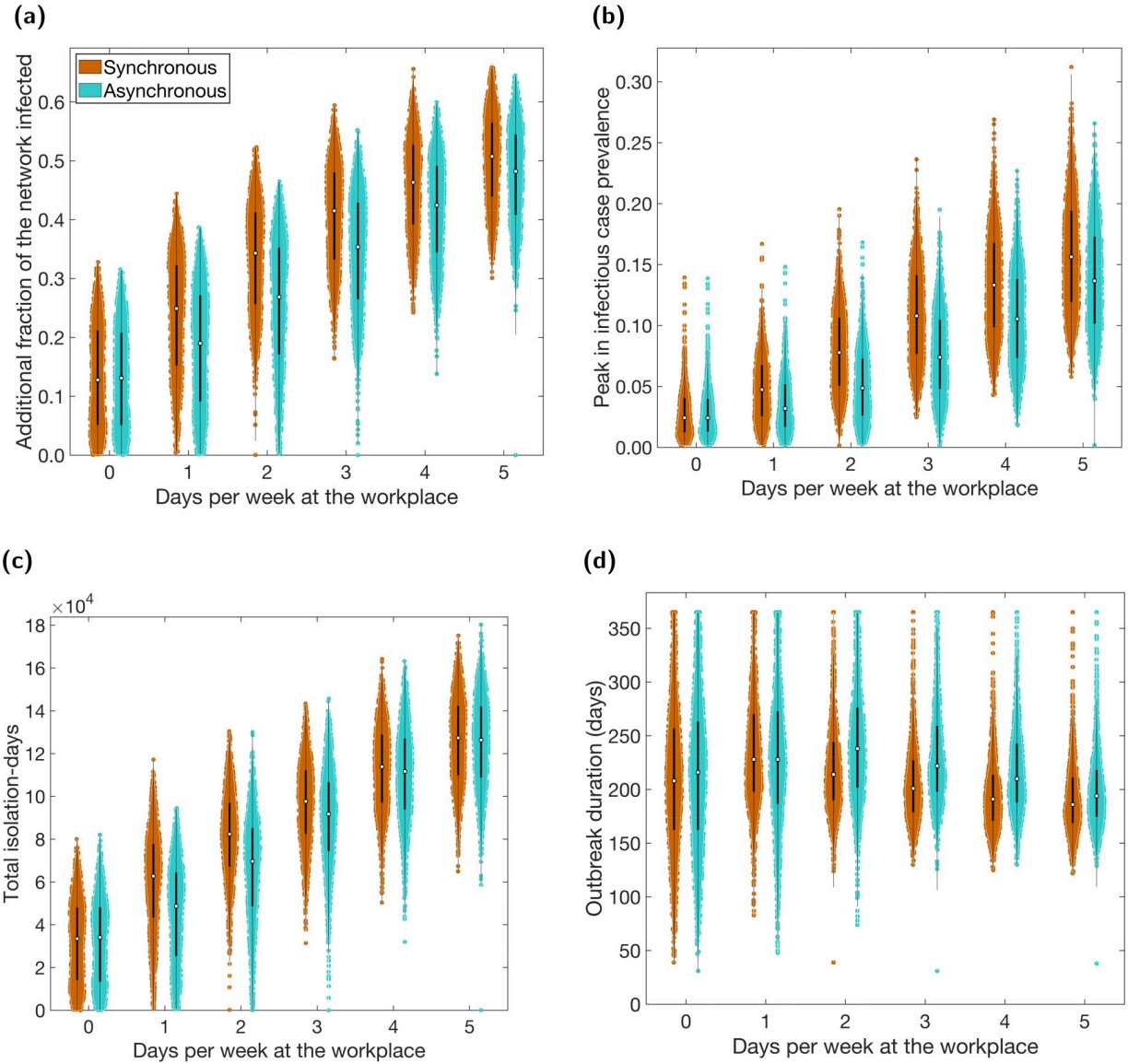

**Fig 3. Case and isolation summary statistics under differing worker patterns.** We introduced NPIs from day 15 onwards and tested synchronous (brown) and asynchronous (cyan) worker patterns, for a range of days spent at the workplace. In all panels, we summarise outputs from 1,000 simulations (with 20 runs per network, for 50 network realisations). We assumed an adherence of 70% in all runs. The white markers denote medians and solid black lines span the 25th to 75th percentiles. We give central and 95% prediction intervals in Table E in the S1 Text. **(a)** Additional proportion of the population that were infectious post introduction of NPIs (day 15 onwards). **(b)** Peak in infectious case prevalence. **(c)** Total isolation-days. **(d)** Outbreak duration (days).

day 15) and the latter 42% (95% PI: 25%–55%) if worker patterns were synchronous, or 35% (95% PI: 15%–51%) if they were asynchronous.

When using asynchronous work patterns we observed fewer total infections, a reduced expected peak prevalence and fewer total isolation-days compared to when using synchronous work patterns (Fig 3(a)–3(c)). These differences depended on the number of days the workers attended the workplace. For total infections and days spent in isolation, the difference between synchronous and asynchronous worker patterns was most pronounced when fewer days were spent at the workplace (assuming this was non-zero). For peak sizes, the same was true for 2–4

days spent at the workplace, with differences diminishing at the extremes. Finally, we found that asynchronous working schedules tended to result in longer outbreaks than synchronous (Fig 3(d)). Note that, for 0 days per week spent at the workplace, synchronous and asynchronous schedules are theoretically identical, with variation between the two caused by stochasticity alone.

## COVID-secure workplaces

We assessed the impact of all workplaces undergoing changes to their contact structures, combined with a possible reduction in transmission risk across workplace contacts (Fig 4).

Without a reduction in transmission risk, we found that restricting workers to teams of up to 10 people was sufficient to reduce the total number infected and size of the infectious case peak (purple violins; Fig 4(a) and 4(b)). Although we did not include team sizes greater than 10, based on the relationship observed in Fig 4(a), we extrapolate that greater team sizes may cause an increase in infections overall. This is likely due to the overall increase in the average number of contacts per worker compared to a non-COVID-secure context, caused by fully connected teams. Intuitively, smaller team sizes resulted in fewer infections. We observed a similar relationship with total isolation-days, although at a team size of 10, isolation-days increased in comparison to a non-COVID-secure context (Fig 4(c)). In contrast, the introduction of teams of workers increased the duration of the outbreak, with smaller teams causing longer outbreaks (Fig 4(d)).

If the risk of transmission was also reduced through COVID-secure measures, we observed further reductions in total infections, the peak in infectious case prevalence and isolation-days (colours; Fig 4(a)–4(c)). The relationship with duration was non-monotonic, with a transmission risk scaling of 1 or 0.75 (no reduction in transmission risk or a 25% reduction in transmission risk) resulting in an increase in duration, but greater reductions (transmission risk scalings of 0.5 and 0.25) resulting in a decrease (Fig 4(d)). However, compared to a non-COVID-secure context, duration was increased in all tested scenarios. Finally, a reduction in transmission risk resulted in more significant changes across all metrics compared to a reduction in team size.

These relationships were reflected in the temporal profiles of the number of infectious and isolating individuals (S1 Text, Figs L and M). As team sizes decreased (right to left), we observed a slight flattening and lengthening of the curves for both metrics. A similar, but more pronounced, effect was seen for decreasing transmission risk (bottom to top).

## Adherence to isolation guidelines and engagement with test-and-trace

Finally, we assessed the sensitivity of our model set-up to different levels of adherence. This applied to both the adherence to isolation measures and engagement with test-and-trace.

We found that increased adherence to isolation and test-and-trace measures resulted in fewer infections overall and a lower peak (Fig 5(a) and 5(b)). From 0% adherence (effectively no NPIs) to 100%, we saw a 50% reduction in overall outbreak size and 75% reduction in infectious prevalence peak size. Nonetheless, whilst increasing adherence introduced greater variability in the final epidemic size (lengthening violin plots with increasing adherence probability), it simultaneously caused a reduction in the variation in the peak in infectious cases (shorter violin plots with increasing adherence probability).

In contrast, higher adherence caused an increase in both the total number of days spent in isolation and the duration of the outbreak (Fig 5(c) and 5(d)). Variability in both these metrics also increased significantly at higher levels of adherence.

These relationships can also be observed in the temporal profiles of infectious cases, isolating individuals and $R_t$ (Fig N in the S1 Text). Increased adherence resulted in a faster decline

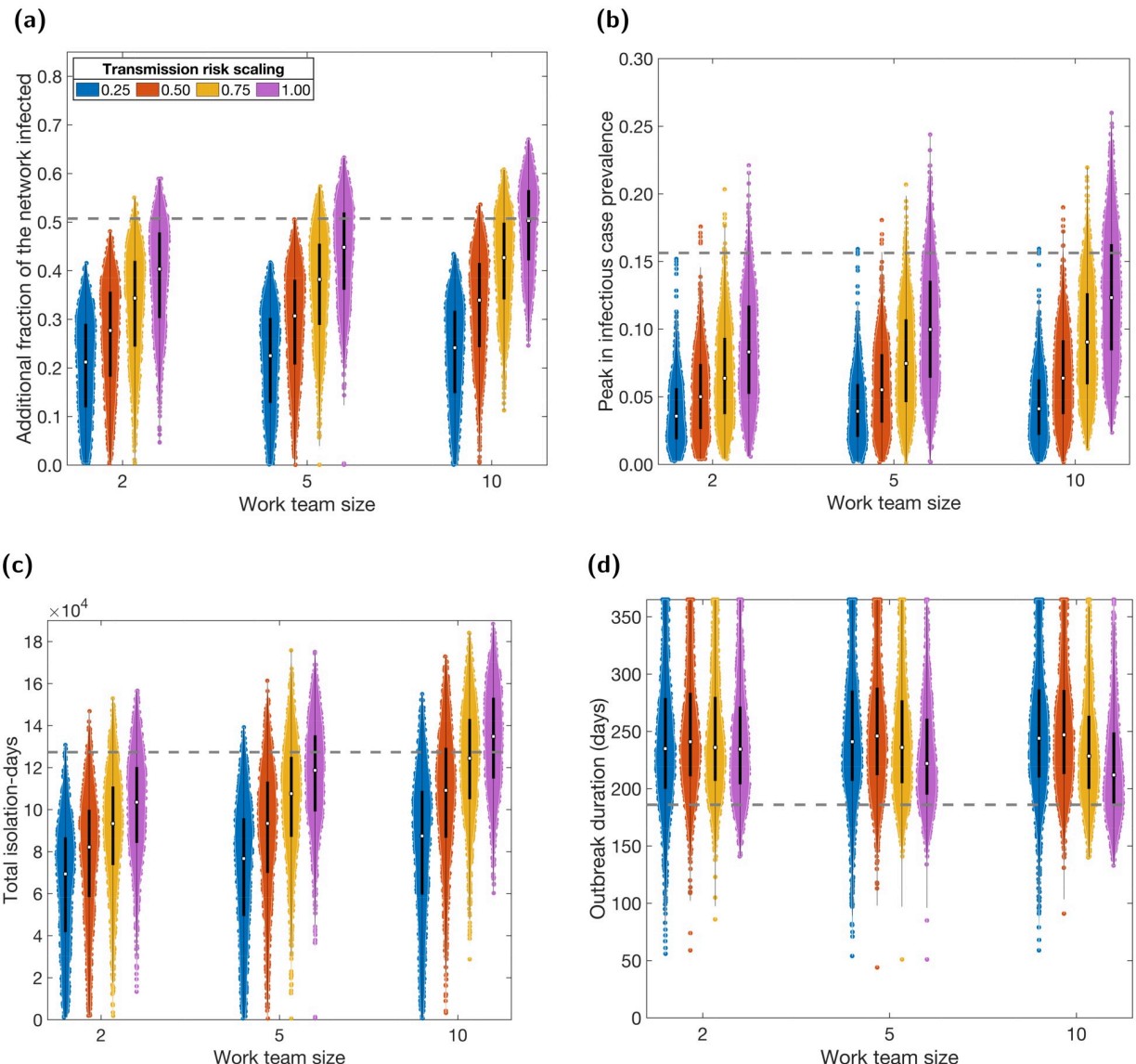

**Fig 4. Case and isolation summary statistics under COVID-secure workplace measures.** We introduced NPIs from day 15 onwards, alongside COVID-secure workplace measures. We tested sensitivity to the maximum work team size (2, 5 or 10) and to the relative scaling of transmission risk under COVID-secure conditions: 0.25 (blue violins), 0.50 (orange violins), 0.75 (yellow violins), 1.00 (purple violins). In all panels, we summarise outputs from 1,000 simulations (with 20 runs per network, for 50 network realisations). We assumed an adherence of 70% in all runs. The white markers denote medians and solid black lines span the 25th to 75th percentiles. The grey dashed horizontal line corresponds to the median estimate with no active COVID-secure workplace interventions. **(a)** Additional proportion of the population that were infectious post introduction of NPIs (day 15 onwards). **(b)** Peak in infectious case prevalence. **(c)** Total isolation-days. **(d)** Outbreak duration (days). We give central and 95% prediction intervals for each summary statistic distribution in Table F in the S1 Text.

in $R_t$ at early stages. However, $R_t$ remains higher during later stages (but still below 1) due to a greater proportion of the population remaining susceptible, causing the temporal profiles of $R_t$ for different adherence levels to cross over. This caused a flatter, longer outbreak, resulting in fewer infections but longer duration. Finally, increased adherence resulted in greater amounts of isolation throughout.

To give an indication of the sensitivity of workplace interventions to adherence, we tested the implementation of COVID-secure workplaces with adherence probabilities of 0

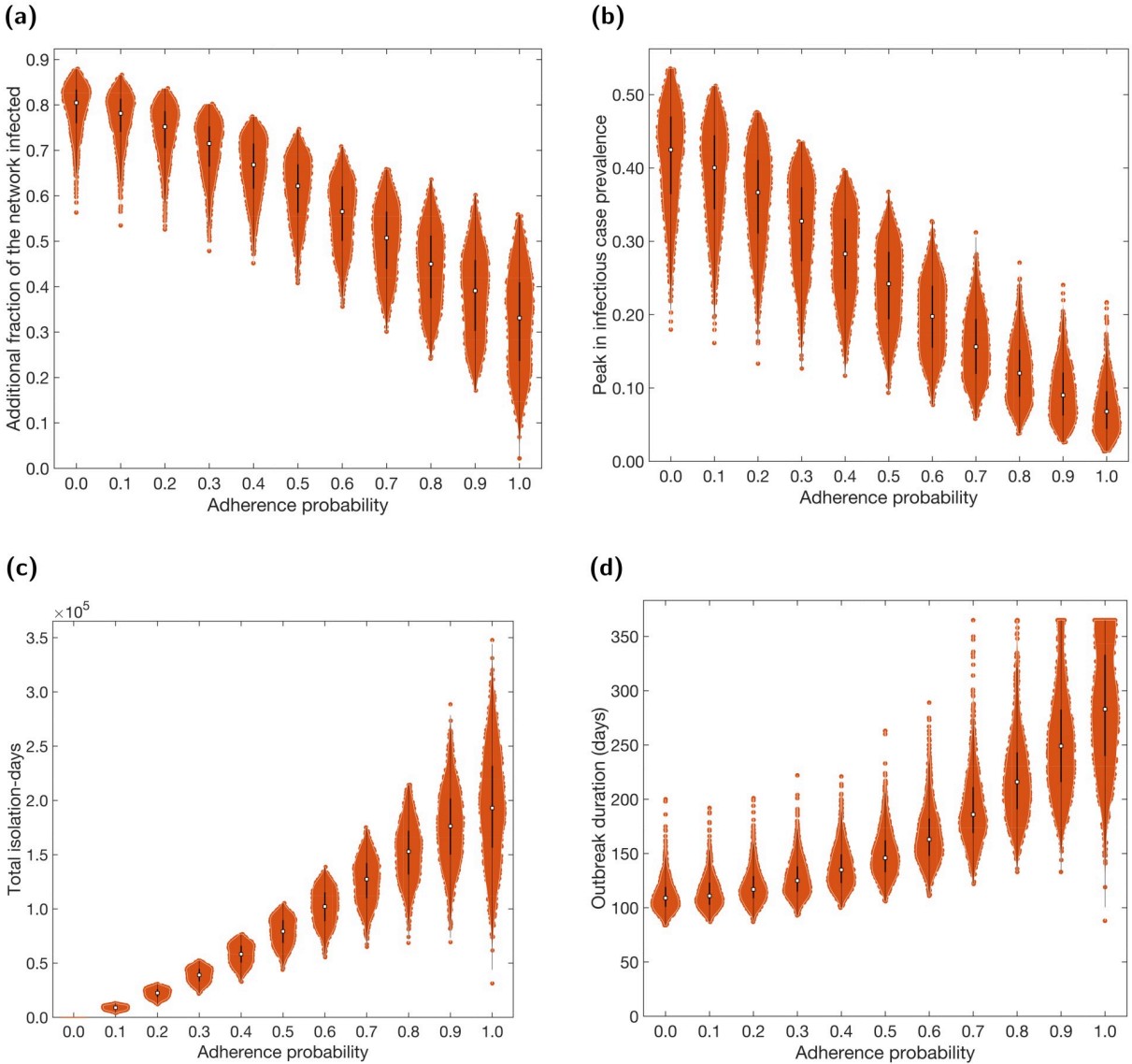

**Fig 5. Case and isolation summary statistics under differing levels of adherence to NPIs.** We introduced NPIs from day 15 onwards, with varying levels of adherence. In all panels, outputs are summarised from 1,000 simulations (with 20 runs per network, for 50 network realisations). The white markers denote medians and solid black lines span the 25th to 75th percentiles. We give central and 95% prediction intervals in Table D in the S1 Text. **(a)** Additional proportion of the population that were infectious post introduction of NPIs (day 15 onwards). **(b)** Peak in infectious case prevalence. **(c)** Total isolation-days. **(d)** Outbreak duration (days).

(equivalently, without isolation and test-and-trace), 0.7 (the default adherence probability), and 1 (all individuals adherent). We present the results for a fixed team size of 5, with varying transmission risk and the three levels of adherence (Fig 6(a)–6(c)), as well as for a fixed scaling of transmission risk of 0.5, with varying team size and the three levels of adherence (Fig 6(d)–6(f)).

We observed that, on average, a lower underlying level of adherence (lighter colours) diminished the relative effectiveness of a workplace targeted intervention at reducing total infections and peak size (Fig 6(a), 6(b), 6(d) and 6(e)). This reduction in median relative effectiveness was generally more pronounced for more intensive interventions (smaller team sizes

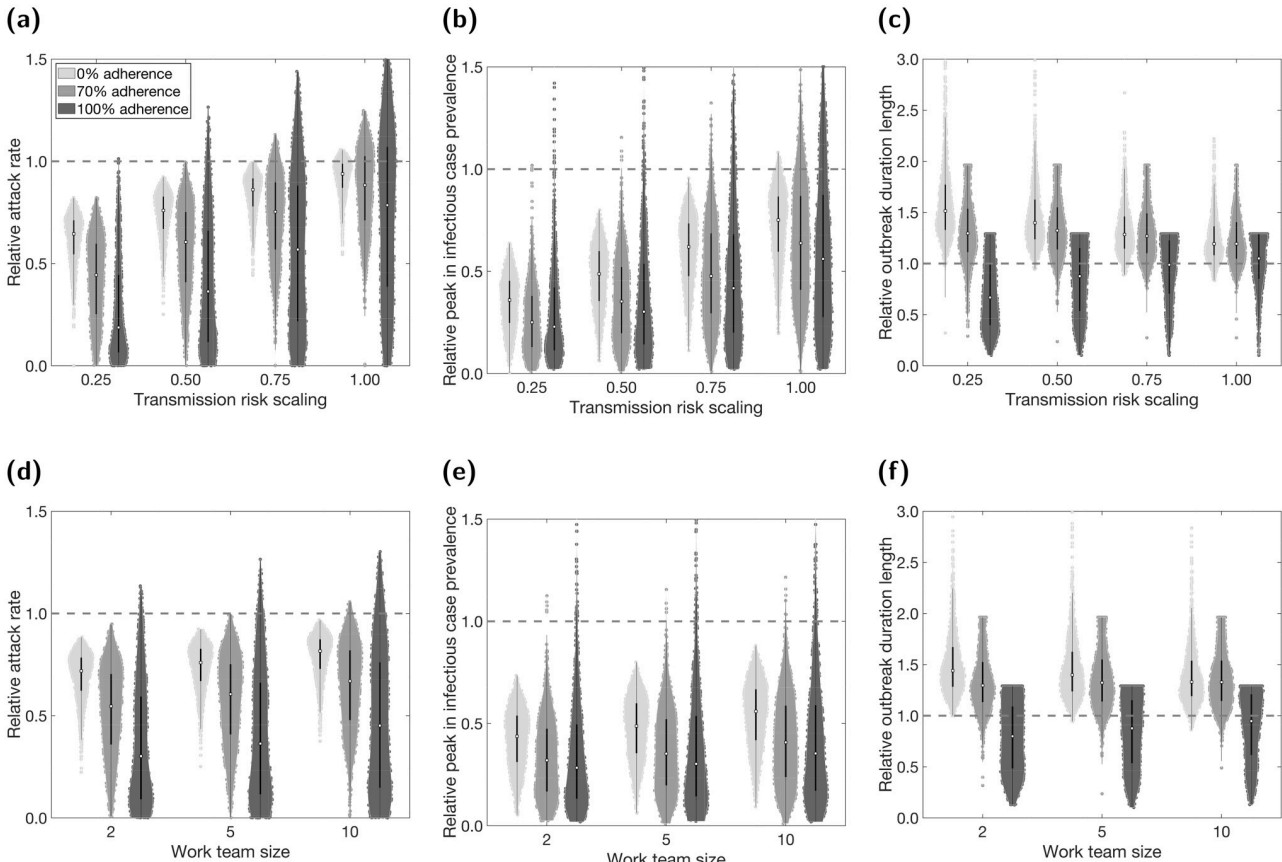

**Fig 6. COVID-secure workplace measures and sensitivity of epidemiological quantities to adherence.** We introduced NPIs from day 15 onwards, alongside COVID-secure workplace measures. We compare three scenarios of adherence to isolation and test-and-trace measures: 0% (lightest shaded violins); 70% adherence (moderate shaded violins); 100% (darkest shaded violins). In panels **(a-c)** we fixed the work team size at 5 and varied the relative scaling of transmission risk under COVID-secure conditions. In panels **(d-f)** we fixed the relative scaling of transmission risk at 0.5 and varied the work team size. We summarise outputs from 1,000 simulations (with 20 runs per network, for 50 network realisations). The white markers denote medians and solid black lines span the 25th to 75th percentiles. The dashed horizontal line corresponds to where estimates from simulations including COVID-secure measures match estimates from simulations that had no active COVID-secure interventions. We present the following summary statistics: **(a,d)** additional proportion of the population that were infectious post introduction of NPIs (day 15 onwards); **(b,e)** peak in infectious case prevalence; **(c,f)** outbreak duration, where we note that some of the violin plots are flat topped, caused by the outbreak duration in any single run being unable to exceed the simulated time horizon of 365 days. We give central and 95% prediction intervals for each summary statistic distribution in Tables G and H in the S1 Text. For the distributions of absolute values for these scenarios, see Fig O and Tables G and H in the S1 Text.

and greater reduction in transmission risk). Lower adherence also reduced the relative variability between simulations in these two metrics.

We observed the opposite effect on outbreak duration (Fig 6(c) and 6(f)): lower adherence caused a relatively greater increase in outbreak duration from the implementation of workplace targeted interventions. Again, this was most pronounced for more intensive interventions. The relative variability in duration appeared to increase with lowered adherence. However, we note that the duration without intervention (dashed lines) is significantly shorter when adherence is lower, thus we are less likely to reach the upper bound for outbreak duration of 365 days (Figs O and P in S1 Text).

Overall, a lack of adherence to underlying isolation and test-and-trace measures led to larger (although shorter duration) outbreaks and worsened the relative performance of workplace interventions. We obtained qualitatively similar relationships between adherence and

the effectiveness of the other workplace targeted interventions considered in this paper (Figs Q-S in S1 Text).

## Discussion

In this study, we have developed a model to analyse the spread of SARS-CoV-2 in the working population, considering the risk of spread in workplaces, households, social and other settings. We have investigated the impact of working from home, asynchronous working patterns, and COVID-secure measures upon disease spread and the time spent in isolation by the working population.

In the UK, an instruction to work from home where possible formed part of a collection of measures that were effective in forcing the initial wave of SARS-CoV-2 infection into decline [35]. Our work supports this effect, finding that requiring a proportion of the population to work from home was effective in reducing the final size of the outbreak and total isolation-days. Under our modelling assumptions and default parameter values, we found a 60–70% decrease in the median estimates of infections, peak infectious prevalence and total isolation-days with everyone working remotely compared to everyone attending the workplace Monday-Friday. However, flattening the epidemic curve in this way would typically result in a pro-longed outbreak duration compared to a scenario without workplace targeted interventions. Furthermore, we demonstrated that a non-uniform proportion working from home across different industry sectors can affect the efficacy of this intervention, even if the overall proportion remains the same. In particular, if sectors with a greater number of dynamic contacts (e.g. hospitality) are also less able to function with workers at home, this could hinder the effectiveness of this intervention. Nonetheless, a sector-specific approach may be explored to determine optimal combinations of work from home percentage across applicable sectors (where working from home is possible), whilst maximising the overall proportion of workers able to attend the workplace.

Another approach to modify work-associated mixing patterns is to alter the scheduling of when workers attend the workplace. We observed (under our default parameter set) up to 20% fewer infections and up to a 40% lower infection peak when using an asynchronous work schedule rather than a synchronous work schedule. These differences between worker pattern implementations were most pronounced when fewer (but non-zero) days were spent at the workplace. We postulate similar outcomes for flexible start and finish times that suit an employee's needs. There are also indications that some businesses envisage retaining flexible working habits longer-term [36], incorporating flexible work times and working from home [37]. This may result in the percentage of the UK workforce reporting a flexible working pattern increasing above a October-December 2019 estimate of 28.5% [38].

It is clear that not all work sectors would be able to implement a work from home policy or allow flexible, asynchronous work patterns. In April, during the first wave of infection in the UK, 46.6% of respondents to a UK-based survey reported having done any work from home in the reference week [39]. However, we have shown that the introduction of COVID-secure measures in the workplace that reduce the number and transmission risk of contacts between workers can help to stem the spread of the virus in the population.

The use of these workplace-targeted interventions should be carefully considered, and the effect and fallout from each weighed against each other. Every decision has an impact on people's lives and livelihoods. In the event of enforced alterations to working practices, it is vital to consider harms to businesses and to personal well-being and mental health, with those affected being fully supported. We believe that a sector-specific combination of workplace-targeted

policies could help to both slow the spread of SARS-CoV-2 and reduce the negative impact to workers, as well as the people and businesses that depend on them.

Prior modelling studies have indicated that nationally applied NPIs (such as social distancing, self-isolation upon symptom onset and household quarantine) may reduce the spread of SARS-CoV-2 [40–42]. Our analysis corroborates these findings, demonstrating that increased adherence to isolation and test-and-trace measures can significantly reduce the size of an outbreak. However, conversely, lower adherence not only worsens the outbreak, but can also reduce the relative effectiveness of workplace targeted interventions. The true adherence of the population, and how this could change over time, should be carefully considered when interpreting these results and applying them in other contexts.

The success of contact tracing operations is not only dependent on engagement from the population, but also on the rapid detection of cases and isolation of contacts (for simplicity we applied a consistent two day turnaround time for this process, though there is observed non-uniformity and temporal variation in these distributions [43]). Given the burden when tracing large numbers of contacts, there is the potential the system could be overwhelmed when the incidence of new cases occurs at a rapid rate [44]. Other operational considerations include the adoption of digital approaches to enable the application of tracing at scale [45]. From the policy maker perspective there are, therefore, trade-offs to consider between investing costs and care in designing sophisticated monitoring networks to enhance rapid detection, or allocating finite resources to alternative interventions as part of the overall package of infectious disease control measures. There would be merits in the use of a coupled transmission model and health economic analysis to determine under what circumstances sophisticated contact tracing systems would be most efficient.

Our evaluations using multiple simulations have incorporated network and epidemiological uncertainty. That being said, there were network, epidemiological and intervention parameters that we assumed fixed and did not vary. In our network contact parameters, we fixed the probability of making contact with an individual in another workplace, compared to an individual within the same workplace, as 0.05 for all work sectors. Sector-specific values would lead to disparities in the amount of clustering between sectors, with higher values increasing the likelihood of multiple workplaces in a sector having cases (less clustering). Additionally, we did not explore uncertainty in the underlying degree distributions for the contact networks. Our conjecture would be inclusion of such uncertainty would increase variability in outcomes. The contact distributions were also informed from a single data source, and it is possible that contact patterns may have changed in the intervening time since the contact survey was undertaken (approximately 10 years). However, contact studies with the richness of data to parameterise work sector contacts are infrequent, thus we have used the most recent data of the required quality available to us.

Another item of prospective sensitivity analysis pertaining to network structure is having a representative proportion of part time workers, though this requires additional assumptions on contact patterns during non-workday weekdays and adds complexity to the network generation. Hence, we have presented a pragmatic approach where we have sought broad insights from what we acknowledge is a simplification of a complex real-world system. Using the analysis we performed around worker patterns, we can postulate that replacing a proportion of the full-time working population with asynchronous, part-time workers would result in a reduced outbreak size and severity.

As part of our epidemiological parameters, we assumed the absolute infectiousness of an asymptomatic case to be less than a symptomatic case, but the duration of infectiousness to be equal. Recent data suggests that while symptomatic and asymptomatic individuals have similar average peak viral loads and proliferation stage durations, their average duration of clearance

stages have been observed to differ [46, 47]. Furthermore, our intervention parameters included a fixed delay in receiving a test result of two days and (for most analyses) a 70% adherence assumption. Our findings may be sensitive to alternative epidemiological model structures and intervention assumptions, with this being a direction of further study.

Our data-driven approach to parameterise the work sector populations and contact structures highlights the heterogeneities that are present in the system. Our work has shown that changing workplace interactions can make a difference to disease transmission and outbreak size, suggesting that relative effectiveness of these factors could contribute to regional variations in epidemiological outcomes. However, there are characteristics of the underlying contact structure that our model formulation does not presently capture. We have not considered clustering of individuals within an individual workplace to capture the fact that, for example, individuals who share an office will be exposed to higher risk. We would expect this to have a stronger effect upon transmission within larger workplaces. In addition, the risk of contracting COVID-19 at work, and the risk of developing serious or fatal COVID-19 should infection occur, will also depend on personal vulnerability [48]. Strong determinants of individual risk are the presence of comorbidities and age, which could be correlated with job type.

Furthermore, our system contained active workers only, with children and the elderly not present. The susceptibility to infection and severity of clinical outcomes generally differs in the youngest and eldest ages compared to those of adults. Within multi-generational households, the relative amount of contact between each generation may differ. Our assumption of members of each household forming a fully connected network could be too general in these circumstances, with an alternative parameterisation required. The impact of age-specific interventions on contact structures also requires attention, such as children switching from attending school in-person to online learning (or vice versa). Thus, the incorporation of age and risk stratification in an expanded network model, and the consequential impact of the disease dynamics amongst the population, merits further investigation.

Another aspect we have not included here is the presence of other respiratory infections. Such an extension would permit the study of test capacity requirements when levels of cough and fever are high due to non-COVID-19 causes. This is especially of concern during the winter period, with expectations of the national test and trace system being put under extra strain [49].

Lastly, while we have informed our model based on UK data, the model may be applied to other countries given the availability of the necessary data to parameterise the model. Modifying the framework to other contexts that have contacts occurring across several reasonably well-defined settings (such as school communities) we perceive as another viable extension.

Models of infectious disease transmission are one tool that can assess the impact of options seeking to control a disease outbreak. Here, we have presented a network model to study epidemic spread of SARS-CoV-2 amongst a population with layered contacts capturing multiple encounter settings, including distinct work sectors. Our work demonstrates the potential uses of this choice of model framework in generating a range of epidemiological measures, which may be analysed to assess the impact of interventions targeting the workforce.

## Supporting information

**S1 Text. Expanded descriptions of the network generation, the network paramterisation, contact risk parameterisation and non-intervention scenario calibration.** Also includes additional figures and tables.
(PDF)

## Author Contributions

**Conceptualization:** Edward M. Hill, Benjamin D. Atkins, Matt J. Keeling, Louise Dyson, Michael J. Tildesley.

**Data curation:** Edward M. Hill, Benjamin D. Atkins, Louise Dyson, Michael J. Tildesley.

**Formal analysis:** Edward M. Hill, Benjamin D. Atkins, Louise Dyson, Michael J. Tildesley.

**Funding acquisition:** Matt J. Keeling, Louise Dyson, Michael J. Tildesley.

**Investigation:** Edward M. Hill, Benjamin D. Atkins, Louise Dyson, Michael J. Tildesley.

**Methodology:** Edward M. Hill, Benjamin D. Atkins, Matt J. Keeling, Louise Dyson, Michael J. Tildesley.

**Software:** Edward M. Hill, Benjamin D. Atkins, Louise Dyson, Michael J. Tildesley.

**Supervision:** Louise Dyson, Michael J. Tildesley.

**Validation:** Edward M. Hill, Benjamin D. Atkins, Louise Dyson, Michael J. Tildesley.

**Visualization:** Edward M. Hill, Benjamin D. Atkins.

**Writing – original draft:** Edward M. Hill, Benjamin D. Atkins.

**Writing – review & editing:** Edward M. Hill, Benjamin D. Atkins, Matt J. Keeling, Louise Dyson, Michael J. Tildesley.

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
