## [Decision Letter · Decision Letter 0]

18 Mar 2021

Dear Dr Hill,

Thank you very much for submitting your manuscript "A network modelling approach to assess non-pharmaceutical disease controls in a worker population: An application to SARS-CoV-2" for consideration at PLOS Computational Biology.

As with all papers reviewed by the journal, your manuscript was reviewed by members of the editorial board and by several independent reviewers. We apologize for the long time of this review process but it proved extremely difficult to find referees and to secure their reviews in a timely fashion.

In light of the reviews (below this email), we would like to invite the resubmission of a significantly-revised version that takes into account the reviewers' comments.  Both referees agree on the value of the work but they indicated a number of important clarifications and improvements in presentation needed to make the results more accessible to readers. We would like to see these addressed.

We cannot make any decision about publication until we have seen the revised manuscript and your response to the reviewers' comments. Your revised manuscript is also likely to be sent to reviewers for further evaluation.

Sincerely,

Mercedes Pascual

Associate Editor

PLOS Computational Biology

Tom Britton

Deputy Editor

PLOS Computational Biology

Reviewer's Responses to Questions

**Comments to the Authors:**

Reviewer #1: This paper presents an analysis of epidemic spread on a network of working adults aged 20-70 years — connected by household, social, workplace and ‘other’ links. Workplaces are stratified into 41 sectors defined by ONS data on UK businesses. Degree distributions for work and social layers of the network are inferred from occupation-specific contact data from the Warwick social contact study. The authors explore the impact of 4 different non-pharmaceutical interventions on the final size, peak incidence, epidemic duration and isolation time for transmission parameters chosen to be yield an initial effective reproduction number of approximately 3, appropriate for wild-type SARS-CoV-2. Results demonstrate that increased worker time spent in the home, asynchronous working patterns over a long working week, reducing work team sizes and increased adherence to contact tracing each reduce outbreak size and flatten the epidemic curve, but do not reduce the duration of the outbreak.

The strengths of this analysis are extensive use of workplace and workplace dependent social contact data to parameterise the network, comprehensive exporation of the uncertainties in outcomes due to parameter uncertainties (but see comments below), and the detail with which interventions are implemented in the model (enabling exploration of nuanced policy options). For these reasons I believe this work represents a novel, valuable addition to existing modelling work on non-pharmaceutical interventions. I have a few suggestions that may aid the clarity of the methodology and interpretation:

1) Can the authors comment on the choice of size of the population modelled (10000 nodes). Is this big enough for the workplace network to be representative of the size distribution and social networks across 41 sectors? Or is the reader to understand that it may not be well sampled, and some of the variation in outcomes is driven by workplace size and network realisation?

2) How does the choice of network size relate to the choice for the distribution of ‘other’ contacts — is this so that other contacts are in approximately the same geographical region?

3) As mentioned care has been taken to convey the range of possible outcomes given uncertainty in transmission parameters, with simulations repeated for repeated for realisations of network and choices of transmission parameters. There are a few parameters that appear may not have be varied. Some of these are noted as subjective choices (e.g. proportion working from home in each sector), and could warrant a sensitivity analysis or further mention in the discussion.

4) Related to the above point — do the authors expect results (particular regarding contact tracing) are sensitive to assumptions regarding the duration of infectiousness of asymptomatic contacts given recent data suggesting it may be the duration of infectiousness rather than absolute infectiousness that differs between asymptomatic and symptomatic cases?

5) Related to points 1) and 3): as I have understood the paper the underlying degree distribution is for the networks assumed to be fixed, and the model does not explore uncertainty in this. Presumably there is uncertainty in inference of the lognormal distributions for number of contacts, and the possibility that contact patterns have changed in the ~10 years since the adotped contact diary survey was undertaken? It would be good to understand whether these uncertainties are dwarfed by other model uncertainties, or discuss the limitations of assuming a fixed underlying network parameterisation.

6) It appears that all workers are assumed to be full time in the default scenario. Have the authors checked whether assuming a representative proportion of part time workers in the baseline scenario changes theh relative benefit of different interventions?

7) Could variation in workplace interactions, proportion able to work from home and proportion isolating when asked, explain local differences in the success of restrictions? Or would this require modelling of other social deterinants of risk as mentioned in the discussion?

Reviewer #2: See the attached document

**Have all data underlying the figures and results presented in the manuscript been provided?**

Reviewer #1: Yes

Reviewer #2: **No: **Data to parametrize the model are taken from other (already published) sources

PLOS authors have the option to publish the peer review history of their article (what does this mean?). If published, this will include your full peer review and any attached files.

Reviewer #1: No

Reviewer #2: No
---

## [Decision Letter · Decision Letter 1]

10 May 2021

Dear Dr Hill,

We are pleased to inform you that your manuscript 'A network modelling approach to assess non-pharmaceutical disease controls in a worker population: An application to SARS-CoV-2' has been provisionally accepted for publication in PLOS Computational Biology.

Best regards,

Mercedes Pascual

Associate Editor

PLOS Computational Biology

Tom Britton

Deputy Editor

PLOS Computational Biology

Reviewer's Responses to Questions

**Comments to the Authors:**

Reviewer #2: Dear Editor,

the authors have reworked quite extensively their manuscript that, in its current form, seems to me equivalently rich, yet more concise and clear. All the points I've raised in the previous round of review have been taken into account and have been satisfactorily answered. In particular, in addition to important clarifications provided in their responses and the appropriate "caveats" included in the Methods section, the authors successfully reorganized the Results section and supported their Discussion with a few (but key) synthetic metrics that give strength to their conclusions.

**Have the authors made all data and (if applicable) computational code underlying the findings in their manuscript fully available?**

Reviewer #2: None

PLOS authors have the option to publish the peer review history of their article (what does this mean?). If published, this will include your full peer review and any attached files.

Reviewer #2: No

---

## [Editor Report · Acceptance letter]

28 May 2021

PCOMPBIOL-D-20-02076R1 

A network modelling approach to assess non-pharmaceutical disease controls in a worker population: An application to SARS-CoV-2

Dear Dr Hill,

I am pleased to inform you that your manuscript has been formally accepted for publication in PLOS Computational Biology. Your manuscript is now with our production department and you will be notified of the publication date in due course.

With kind regards,

Kata Acsay
